# Are Foliar Nutrition Status and Indicators of Oxidative Stress Associated with Tree Defoliation of Four Mediterranean Forest Species?

**DOI:** 10.3390/plants11243484

**Published:** 2022-12-13

**Authors:** Lucija Lovreškov, Ivana Radojčić Redovniković, Ivan Limić, Nenad Potočić, Ivan Seletković, Mia Marušić, Ana Jurinjak Tušek, Tamara Jakovljević, Lukrecija Butorac

**Affiliations:** 1Croatian Forest Research Institute, Cvjetno Naselje 41, 10450 Jastrebarsko, Croatia; 2Faculty of Food Technology and Biotechnology, University of Zagreb, Pierottijeva 6, 10000 Zagreb, Croatia; 3Institute for Adriatic Crops and Karst Reclamation, Put Duilova 11, 21000 Split, Croatia

**Keywords:** antioxidative enzymes, chlorophyll, defoliated trees, hydrogen peroxide, lipid peroxidation, nutrient concentration, oxidative stress, *Pinus* spp., *Quercus* spp., undefoliated trees

## Abstract

Mediterranean forest ecosystems in Croatia are of very high significance because of the ecological functions they provide. This region is highly sensitive to abiotic stresses such as air pollution, high sunlight, and high temperatures alongside dry periods; therefore, it is important to monitor the state of these forest ecosystems and how they respond to these stresses. This study was conducted on trees in situ and focused on the four most important forest species in the Mediterranean region in Croatia: pubescent oak (*Quercus pubescens* Willd.), holm oak (*Quercus ilex* L.), Aleppo pine (*Pinus halepensis* Mill.) and black pine (*Pinus nigra* J. F. Arnold.). Trees were selected and divided into two groups: trees with defoliation of >25% (defoliated) and trees with defoliation of ≤25% (undefoliated). Leaves and needles were collected from selected trees. Differences in chlorophyll content, hydrogen peroxide content, lipid peroxidation and enzyme activity (superoxide dismutase, catalase, ascorbate peroxidase, non-specific peroxidase), and nutrient content between the defoliated and undefoliated trees of the examined species were determined. The results showed that there were significant differences for all species between the defoliated and undefoliated trees for at least one of the examined parameters. A principal component analysis showed that the enzyme ascorbate peroxidase can be an indicator of oxidative stress caused by ozone. By using oxidative stress indicators, it is possible to determine whether the trees are under stress even before visual damage occurs.

## 1. Introduction

Various abiotic stresses such as ground-level ozone (O_3_), intense drought, and acid compounds strongly limit the growth of plants in the Mediterranean region [1,2]. These ecosystems are affected by the combined influence of local, regional, and long-distance pollution caused by human activities [3,4,5,6,7,8]. Additionally, these factors tend to be the main cause of the decline in forest vitality and productivity [2,9,10,11,12]. For example, abiotic stress caused by ground-level ozone, which is phytotoxic to plants and harmful to humans, can cause chlorosis and necrosis [13,14]. Additionally, nitrogen deposition can have a positive influence on forest productivity but can also cause forest degradation through acidification and nutrient deficiency. Changes to the frequency and intensity of climatic extremes (e.g., heat waves, precipitation, and storms) are also among the various factors that can have a serious impact on forest health and vitality. [4,15,16]. Proper forest monitoring is therefore essential to document their conditions and to investigate the effects of—and relationship with—stress factors [17].

Crown defoliation is a commonly used indicator of acute stress that is defined as the loss of leaves in the assessable crown compared to a reference tree, and it is observed regardless of the cause of foliage loss [18,19]. In field conditions, it is difficult to confirm if defoliation is a consequence of one particular stress factor because many factors simultaneously influence the tree balance [20]. Plants adapt to survive, and the allocation of resources (nutrients) at the cost of losing leaves is one of the known plant defense mechanisms [21]. However, leaf loss is still used as a health indicator in Europe according to the methodology of the International Co-operative Program on Assessment and Monitoring of Air Pollution Effects on Forests (ICP Forests) [18]. One of the main factors of forest health and functioning is the circulation of nutrients between the soil and plants [22]. The concentration of biogenic elements and their relationships in the leaves of forest trees are important indicators of their functioning and provide insight into the state of nutrition. Macronutrients such as nitrogen (N), phosphorus (P), potassium (K), calcium (Ca), and magnesium (Mg) are important parts of plant metabolism. Nitrogen is one of the most important macronutrients required by plants. All proteins that are part of the structure of chlorophyll consist of nitrogen-containing amino acids [23]. Due to its role in the production of chlorophyll and specific proteins such as ribulose-1,5-bisphosphate carboxylase (RuBisCO), which is responsible for the uptake of CO_2_, nitrogen is a crucial element for the development of the photosynthetic process and vegetative growth of a plant [23]. Nitrogen deficiency can decrease the photosynthetic activity and longevity of leaves. Generally, nutrient deficiency, a source of abiotic stress, can cause increased hydrogen peroxide (H_2_O_2_) and lipid peroxidation (LPO) depending on the plant species [24]. Enzyme activity tends to increase as a result of increased H_2_O_2_ and LPO [24]. Macronutrient deficiencies disrupt plant metabolism and functions including various physiological or metabolic activities that increase the production of reactive oxygen species (ROS) in organisms. The excess production of ROS causes oxidative stress in plants [24]. The lack of these nutrients generally manifests as chlorosis, necrosis, defoliation, and lower growth and productivity, which can eventually lead to plant death [18,25,26,27]. In such conditions, plant defense mechanisms play a very important role, especially in maintaining the normal functionality and survival of a plant [28]. Changes in the environment trigger plant defense mechanisms, which remove the undesirable products created because of oxidative stress. These products are called ROS, and their production is a consequence of aerobic metabolism [29]. Accumulation of ROS can cause cell damage such as lipid peroxidation (LPO), protein damage, and membrane destruction leading to cell death. These effects manifest themselves in various ways such as leaf yellowing, chlorosis, and necrosis [29,30]. To prevent the damage caused by the oxidation of essential molecules (proteins, DNA, RNA, and lipids) inside a cell, the ROS are neutralized by special molecules, enzymes, and antioxidants. Antioxidant enzymes include superoxide dismutase (SOD), ascorbate peroxidase (APX), catalase (CAT), and peroxidase (POD), which are part of the defense mechanism for removing ROS [25]. The first line of defense is SOD which converts superoxide (O_2_^−^) to H_2_O_2_, whereas POD, CAT, and APX decompose H_2_O_2_ to water (H_2_O) [31,32]. The biochemical response of the plant depends on the environment in which it grows, the sources of stress that affect it, and the type of plant [33].

Considering various sources of abiotic stress and the fragility of Mediterranean ecosystems, there is an increasing need to improve the knowledge of Mediterranean forest ecosystems, especially in Croatia where few epidemiological studies have been conducted under Mediterranean field conditions [7,8,34]. This study was focused on four of the most widespread and most important forest tree species of the Mediterranean region: pubescent oak (*Quercus pubescens* Willd.), holm oak (*Quercus ilex* L.), Aleppo pine (*Pinus halepensis* Mill.), and black pine (*Pinus nigra* J. F. Arnold.). We conducted field observations and visual assessment of crown condition (defoliation) and complemented these observations with measurements of stress indicators, including chlorophyll content (Chl), H_2_O_2_, LPO, and antioxidative enzyme activities, including those of SOD, CAT, POD, and APX. Our aim was to test whether foliar nutrition status and the indicators of oxidative stress were associated with tree defoliation. Differences in foliar nutrition status and oxidative stress indicators between trees with defoliation of >25% (defoliated) and trees with defoliation of ≤25% (undefoliated) were tested. Furthermore, using a principal component analysis (PCA) we investigated whether the oxidative stress indicators were associated with defoliation and environmental variables to determine if they could be used as early indicators of forest tree health.

## 2. Results and Discussion

### 2.1. Foliar Nutrition Status of Four Selected Mediterranean Species

A common problem occurring in the Mediterranean region is nutrient deficiency [35]. Nutrient deficiency is considered an abiotic stress because it can cause irregular plant development, which leads to (among other things) defoliation [36]. For example, Ferretti et al. [37] determined that common beech trees with crown defoliation over 25% have an imbalance of N with K and Ca. Thus, to gain insight into the nutrient status of our examined species, the concentration of nutrients (N, P, K, Ca, and Mg) was determined for selected undefoliated and defoliated trees.

The nutrient concentrations (N, P, K, Ca, and Mg) in the leaves and needles of pubescent oak, holm oak, Aleppo pine, and black pine are shown in Table 1. Plant nutrition was generally in an optimal range with a few exceptions [38]. As expected, we identified significant differences in nutrient concentrations among species. However, regarding holm oak (K) and Aleppo pine trees (Ca and Mg), the nutrient status of the defoliated trees was in the low nutritional range (except for undefoliated trees for Mg) (Table 1).

A nutrient imbalance can occur due to various external influences on plants [10,36,39]. One of the most phytotoxic abiotic factors today is ozone [10,40]. Nitrogen plays an important role in regulating plant sensitivity to ozone; i.e., with a sufficient amount of the N nutrient, plants can neutralize higher amounts of ozone without causing damage to the plant tissue [41]. The addition of P can cause an increase in tree biomass and an increase in ozone tolerance but only when the amount of nitrogen is low [41]. A study in northern Spain reported that P was higher in more defoliated beech trees [42]. In our study, a significantly higher concentration of P was found in the defoliated pubescent oak trees and black pine trees compared to the undefoliated trees. Furthermore, a previous study on sampled leaves of the Persian oak found that the concentrations of Ca increased with increased drought [43]. Potočić et al. [44] found that drought caused low needle Ca concentrations in Silver fir trees of all defoliation classes. Calcium and K have a common role in the regulation of water in trees [45]. During the dry period, cell division and leaf growth are reduced due to reduced cell turgor. There is also a decrease in the rate of photosynthesis because the intake of CO_2_ is limited during drought conditions, which subsequently limits the growth of leaves [43]. Thus, due to the increase in defoliation, the remaining green leaves of trees in categories with higher degrees of defoliation must absorb a greater amount of Ca to perform vital activities such as photosynthesis. In our study, this same process could be observed in the undefoliated pubescent oak and Aleppo pine trees as a defense mechanism. It is possible that the trees may have also been under stress due to environmental conditions in the Mediterranean region, such as high concentrations of ozone and a large amount of nitrogenous and acidic compounds [7,8,34]. Although the results showed a mostly optimal concentration of nutrients (Table 1), various factors affect the resistance of the examined species, and the concentration of nutrients could not be considered as a potential cause of stress in the tested species.

### 2.2. Content of Chl, H_2_O_2_, and MDA in Four Selected Mediterranean Species

Chlorophyll content depends on various sources of abiotic and biotic stresses, such as ozone, atmospheric and soil pollution [46,47], stress intensity and duration [39,46], and plant nutritional status, especially the status of nitrogen as a major component of chlorophyll [23,48]. In our study, the content of chlorophyll a (Chl-a), chlorophyll b (Chl-b) and total chlorophyll (Chl-tot) was determined (Table 2). Different contents of chlorophyll were identified, depending on the species. In oak species, a significantly higher Chl-tot content was found in the undefoliated trees, whereas in black pine, significantly more Chl-tot was found in the defoliated trees. For Chl-a, significant differences were found between the defoliated and undefoliated trees for pubescent oak, holm oak, and black pine. For Chl-b, significant differences were found for holm oak and Aleppo pine. Higher Chl-a, Chl-b, and Chl-tot contents were found in the undefoliated oak trees. However, higher Chl-a, Chl-b, and Chl-tot contents were found in pine defoliated trees.

The content of H_2_O_2_ was only significantly different between the defoliated and undefoliated trees for the black pine species. A higher content of H_2_O_2_ was found in the undefoliated trees of all species except for the Aleppo pine trees. Furthermore, significantly different contents of MDA were found for all species except black pine. The results showed a higher MDA content in the undefoliated trees of pubescent oak and black pine but a lower MDA content in the undefoliated trees of holm oak and Aleppo pine (Table 2).

In the literature, there have not been many similar studies conducted in field conditions due to the complex relationship between trees and various environmental variables. Most studies were performed in controlled conditions [33,39,47,49]. However, some of the results obtained in this study can be compared to the previous results obtained under controlled conditions.

Studies have found that the content of photosynthetic pigments is affected by the most influential factors in the Mediterranean region: ozone, nutrient deficiency, and drought [33,39,47]. Drought leads to reductions in chlorophyll, especially if other sources of stress, such as ozone, are involved. Furthermore, it was found that during stressful conditions, deciduous species focused on the synthesis of chlorophyll and the maintenance of photosynthesis to create enough energy for reproduction during the growing season [49]. On the other hand, evergreen species were more focused on preservation, which resulted in lower growth (lower photosynthesis), thus suggesting that their goal was leaf survival (up to two years). Therefore, evergreen species invest more energy in extending leaf life and less energy in chlorophyll formation [49]. In our study, significantly lower contents of Chl-a, Chl-b, and Chl-tot were found in the defoliated trees of the holm oak species (Table 2). For the pine species, a significantly higher content of chlorophyll was found in the defoliated trees (Table 2), suggesting that the pine species were investing energy in chlorophyll synthesis. Pines are known to shed their needles as a defense mechanism to protect themselves from drought and other stresses [3,50]. Due to their lack of leaf surface, pines invested energy into the synthesis of chlorophyll to compensate for this loss. Accordingly, higher chlorophyll content was found in the defoliated trees. Nutrients such as nitrogen (one of constituents of chlorophyll) were not considered to cause differences in chlorophyll concentrations between the defoliated and undefoliated trees, as the N concentrations in both tree groups were in the optimal range (Table 1).

Under stress-free conditions, the H_2_O_2_ that is created inside a cell via cell metabolism serves as a signaling molecule [32]. However, during stressful conditions, H_2_O_2_ is produced in excessive amounts and causes damage inside cells [29]. The H_2_O_2_ contents found in our study are listed in Table 2. Significant differences in H_2_O_2_ contents were determined between the defoliated and undefoliated black pine trees, but the other species did not show any significant differences. Furthermore, H_2_O_2_ is one of the main precursors of LPO. Lipid peroxidation occurs when damage is caused to the cell membrane as a result of intense oxidative stress and insufficient removal of the ROS [29]. As a consequence of this damage to the lipid membrane, MDA is produced [32]. For holm oak and Aleppo pine, a significantly higher content of MDA was found in the defoliated trees than in the undefoliated trees (Table 2). For pubescent oak, a significantly lower content of MDA was found in the undefoliated than in the defoliated trees. For holm oak and Aleppo pine, significantly greater differences in MDA were found in the defoliated trees than in the undefoliated trees. Interestingly, a significant difference in H_2_O_2_ content was only found in black pine. Furthermore, no significant difference in MDA content was found for this species, demonstrating that black pine efficiently removed H_2_O_2_ in both the defoliated and undefoliated trees (Table 2).

A study on holm oak saplings examined the influence of ozone and salinity and demonstrated lipid peroxidation after exposure to the aforementioned sources of stress, revealing significant difference compared with plants that were exposed to one or both sources of stress [2]. Furthermore, the results of research on oak species (*Quercus brantii* Lindl.) showed that an increase in drought stress caused an increase in MDA content and resulted in an increase in Ca content in the leaves [43]. In our study, no significant differences in the content of MDA between the undefoliated and defoliated black pine trees were found, but significant differences were found for all other species. These results indicated that the content of H_2_O_2_ activated defense mechanisms in the undefoliated trees and that these mechanisms were already activated in the defoliated trees, which resulted in no significant differences in the content of MDA. Our recent research has shown that all the examined species are under oxidative stress due to the high ozone concentrations determined on all four plots [7].

### 2.3. Activity of Antioxidative Enzymes in Four Selected Mediterranean Species

For the optimal growth and development of a plant in a changing environment, it is essential to neutralize ROS production [25]. The results of enzyme activity in the leaves and needles of the defoliated and undefoliated trees are shown in Figure 1. Significant differences in SOD activity were only found between the undefoliated and defoliated black pine trees. Higher SOD activity was found in the defoliated black pine trees, whereas SOD activity was lower in the undefoliated black pine trees (Figure 1a). CAT activity showed significant differences between the undefoliated and defoliated trees for all species except for the holm oak. The results showed higher CAT activities in the defoliated holm oak and Aleppo pine trees and vice versa for the pubescent oak and black pine trees (Figure 1b). Significant differences in POD activity were only recorded between the undefoliated and defoliated black pine trees. As shown in Figure 1c, higher POD activities were found in the defoliated holm oak and Aleppo pine trees and vice versa for the pubescent oak and black pine trees. Significant differences in APX activity were found between the undefoliated and defoliated trees for all species except for holm oak. Higher APX activities were determined in the undefoliated trees of all species except the holm oak trees (Figure 1d).

In our study, the significant differences in H_2_O_2_ and SOD activity between the undefoliated and defoliated black pine trees suggest that the main product of ROS is a superoxide radical. Similar conclusions were also drawn during research on the pubescent oak species [33]. The role of the antioxidant enzyme SOD is to directly catalyze superoxide radicals to H_2_O_2_ [33,51]. In our study, the accumulated H_2_O_2_ activated the enzymes CAT, POD, and APX (Figure 1). In a controlled environment, adding ozone to chambers containing the black ash (*Fraxinus ornus* L.) was found to increase the amount of ROS and, consequently, the activity of the SOD and CAT enzymes [52]. In our study, many factors could influence the activation of defense mechanisms in the examined species, such as nitrogen and acidic compounds of atmospheric deposition, high ozone concentrations, and high temperatures [7,8].

According to the obtained results, only the activation of the APX enzyme could be considered an indicator of stress for the examined species. This suggests that the APX enzyme activity could be used to determine whether a plant is under stress even before visual damage occurs. Therefore, it is important to monitor undefoliated trees and defoliated trees to be able to react in a timely manner with the aim of preserving these sensitive forest ecosystems.

### 2.4. Relationship between Defoliation, Oxidative Stress Indicators, and Environmental Variables

In our previous investigations, we examined the effects of air pollution on the condition of the forest ecosystem by analyzing tree vitality and obtained interesting results [7,8,34]. For example, higher levels of N, acid compounds, and ozone concentrations were measured in dominant forest species along the Adriatic coast [7,8,34]. The highest percentages of defoliated trees were found for the pubescent oak, Aleppo pine, and black pine plots [8]. Based on these findings, the relationship between defoliation, oxidative stress indicators, and environmental variables was tested using a PCA analysis to gain insight into the behavior of an individual species in response to external influences.

A principal component analysis was carried out to analyze the relationship between the spatial distribution patterns of the indicators of oxidative stress and environmental conditions for the four investigated tree species separated into two groups: defoliated and undefoliated trees (Figure 2). In this analysis, the indicators of oxidative stress included APX, CAT, Chl-a, Chl-b, Chl-tot, H_2_O_2_, MDA, POD, and SOD, whereas the environmental conditions included ozone (O_3_), air temperature (T), air relative humidity (RH), solar radiation (Rad) and soil water content at different depths: SWC1 = 0–7 cm, SWC2 = 7–28 cm, SWC3 = 28–100 and SWC > 100 cm, and rain. For the analyzed combinations of variables, the first two principal components (PCs) accounted for 99.99% of the total variance. In Figure 2, the positive correlation between variables is greater when the variable representative points are close to one another and the circle, whereas the negative correlation is greater when the points are distant from the circle’s center and the circle. The variable representative points are uncorrelated when they are orthogonal to the center of the circle. Our results show that the correlations between stress indicators and antioxidants, environmental conditions, and soil water content differ for the different tree species and defoliation states. APX (which catalyzes the reduction of H_2_O_2_ to water) was found to be positively correlated with SWC1 and SWC2, whereas POD was found to be positively correlated with rain and RH in the undefoliated pubescent oak trees. On the other hand, for the defoliated pubescent oak trees, APX was positively correlated with O_3_ concentration. Furthermore, for the undefoliated holm oak trees, APX was positively correlated with RH, SWC, Rad, and T; for the undefoliated Aleppo pine trees, APX was positively correlated with SWC1, SWC2, and SWC3; and for the undefoliated black pine trees, APX was positively correlated with O_3_ concentration. APX release could be a potential defense mechanism of the selected species against the oxidative stress caused by high ozone concentrations on the examined plots. The separation of SWC4 from the other SWC1-3 variables was due to a difference in water content depending on soil depth and season. In southern Europe, the upper soil layer dries out faster than the deeper layers during the warm and dry seasons [53]. The lack of a clear relationship between the stress indicators and environmental variables of the defoliated and undefoliated trees of the examined species could partially be due to a complex relationship between trees and their surroundings. The PCA revealed that the MDA concentration was positively correlated with O_3_ concentration for the undefoliated pubescent oak, holm oak, and black pine trees. The MDA concentration was also positively correlated with RH and rain for the undefoliated Aleppo pine trees. A plant can reach a state of oxidative stress due to various external influences (biotic and abiotic), and air pollutants are one of the main abiotic factors that cause oxidative stress in forest ecosystems. Therefore, we expected to find differences in the response to oxidative stress because this study was conducted on four different forest species. In this study, the biochemical analysis showed that possible damage could occur. The results of the oxidative stress indicators showed that the examined species were under stress. In field conditions, various sources of stress could cause the activation of defense mechanisms in these four species. Higher chlorophyll content in the undefoliated oak trees suggests an investment of energy into chlorophyll synthesis. More chlorophyll was found in the defoliated pine trees, suggesting the investment of energy into survival. The accumulated H_2_O_2_ caused lipid peroxidation, which was confirmed by the presence of MDA in all species. The principal component analysis identified different responses depending on the tree species and on the defoliated or undefoliated status.

## 3. Materials and Methods

### 3.1. Selected Plots

Measurements were obtained from different Mediterranean forest ecosystems of the Eastern Adriatic coast in four plots. The most common broadleaf species in this part of the Mediterranean are *Quercus pubescens* Wild. (pubescent oak) and *Quercus ilex* L. (holm oak) in the Istria region and *Pinus halepensis* Mill. (Aleppo pine) and *Pinus nigra* J. F. Arnold. (black pine) in the Dalmatia region [54]. A detailed description of the plots can be found in [7,8].

### 3.2. Crown Condition Assessment and Collection of Leaves and Needles

The defoliation status of the selected sample trees was assessed annually by trained and experienced personnel in 2017 and 2018 [18]. After the assessment, trees were divided into two categories: defoliated (defoliation > 25%) and undefoliated (defoliation ≤ 25%) trees. A threshold of 25% is traditionally used to distinguish between “healthy” and “damaged” trees [18,55]. Defoliation of >25% is also reflected by a distinct change in the functional leaf traits and other indicators of tree health [55]. From these two categories, three trees of each species were randomly selected and sampled. Collected leaves and needles were harvested in late summer when the leaves were fully developed. Leaves were harvested off the upper light-exposed portion of the crown at 2–4 m height [18]. Collected samples were used for analyses of nutrient content and oxidative stress indicators.

### 3.3. Foliar Nutrition Analysis

Plant samples used in the foliar nutrition analysis were oven-dried at 80 °C for 24 h. Approximately 0.3 g of sub-samples were digested using a mixture of 30% H_2_O_2_ and 65% HNO_3_ for plants and 0.3 g of the elemental analyzer (LECO CNS-2000, St. Joseph, MI, USA) [56]. The concentration of nutrients was determined as follows: N using an elemental analyzer P with a spectrophotometer (LaboMed UV-VIS, Los Angeles, CA, USA) and Ca, P, Mg, and K using an atomic absorption spectrophotometer (Perkin Elmer AAS AAnalyst 700, Waltham, MA, USA) [56].

### 3.4. Estimation of Chlorophylls

Leaves or needles (0.1 g) were homogenized using a mortar and pestle in 10 mL of chilled 80 % acetone. The chlorophyll extract was refrigerated at 4 °C for 24 h. The samples were then centrifuged at 4000× *g* for 10 min, and the absorbance of the supernatant was measured using a spectrophotometer (LaboMed UV-VIS) at 663 nm and 645 nm. The content of photosynthetic pigments is expressed as µg of photosynthetic pigments per gram of fresh weight of leaves or needles (µg g^−1^ FW) [57].

### 3.5. Evaluation of Lipid Peroxidation (LPO) and Hydrogen Peroxide (H_2_O_2_)

Leaves or needles (0.2 g) were homogenized using a mortar and pestle in 2 mL of 0.1% thiobarbituric acid (TBA). The extracts were centrifuged at 10,000× *g* for 20 min, and the resulting supernatants were used to determine lipid peroxidation and H_2_O_2_.

The level of lipid peroxidation in the leaves and needles of the selected species was evaluated by measuring malondialdehyde (MDA) using the thiobarbituric acid method, which yields a colored product [58]. A mixture of supernatant and 0.5% thiobarbiturate acid (TBA) in 20% trichloroacetic acid (TCA) was heated at 95 °C for 30 min and then cooled in an ice bath. After centrifugation at 10,000× *g* for 10 min, the absorbance of the supernatant was read at 532 nm, and the correction for unspecific turbidity was conducted by subtracting the absorbance at 600 nm on the LaboMed UV-VIS. A total of 0.25% TBA in 10% TCA served as the blank. The MDA content was calculated according to its extinction coefficient of 155 mM^−1^ cm^−1^ and expressed as nmol g^−1^ FW [58,59].

To determine hydrogen peroxide (H_2_O_2_), a 100 mM potassium phosphate buffer and potassium iodide were added to the supernatant. The absorbance of the mixture was read at 390 nm on the LaboMed UV-VIS. The molar extinction coefficient for H_2_O_2_ is 0.28 µM^−1^ cm^−1^, and the amount of H_2_O_2_ is expressed as µmol per gram of fresh weight of leaves or needles (µmol g^−1^ FW) [58,59].

### 3.6. Antioxidant Enzyme Extraction and Assay

For the enzyme analysis, 0.2 g of leaves or needles were homogenized in an ice bath with the addition of polyvinyl polypyrrolidone (PVPP) in a 100 mM potassium phosphate (K_2_HPO_4_/KH_2_PO_4_) buffer solution, pH 7.0, that included 1 mM EDTA and L-ascorbic acid using a pre-chilled mortar and pestle. The homogenates were centrifuged at 15,000× *g* and 4 °C for 20 min. The supernatant was used for the enzyme activity and protein content assays [59]. All analyses were completed using a LaboMed UV-VIS spectrophotometer.

The total soluble protein contents of the enzyme extracts were estimated according to Bradford [60]. A 40 µL sample of the supernatant was added to 1.2 mL of Bradford reagent followed by incubation in a dark room for 15 min. The absorbance of the supernatant was read at 595 nm. The protein content was expressed in mg protein g^−1^ FW.

The activity of superoxide dismutase (SOD) was determined by measuring the inhibition of the photochemical reduction of nitro blue tetrazolium (NBT) using the method of Beauchamp [61]. An aliquot of enzyme extract was added to a reaction mixture containing a 50 mM potassium phosphate (KPO_4_) buffer (pH 7.8), 13 mM methionine, 75 mM NBT, 2 mM riboflavin, and 0.1 mM ethylenediaminetetraacetic acid. The test tubes were shaken, and the enzymatic reaction was started by turning on the 36 W fluorescent lamp. The increase in absorbance due to formazan formation was read at 560 nm. One unit of SOD activity was defined as the amount of enzyme that inhibits the NBT photoreduction by 50%. The activity of SOD is expressed as a unit (U) mg^−1^ protein [59].

The catalase (CAT) activity was determined using the decomposition of H_2_O_2_ and was measured spectrophotometrically [62]. The decrease in absorbance was assessed at 240 nm in a quartz cuvette. The reaction mixture contained 50 mM KPO_4_ buffer (pH 7.0), 10 mM H_2_O_2_, and an enzyme extract. Catalase activity is expressed as U mg^−1^ protein [58,59].

Ascorbate peroxidase (APX) activity was measured according to the method of Ambriović-Ristov et al. [58]. A suitable enzyme extract aliquot was added to the reaction mixture containing 0.5 mM ascorbate and 0.12 mM H_2_O_2_ in a 50 mM phosphate buffer (pH 7.0). APX activity was determined according to the decrease in absorbance of ascorbate at 290 nm (ε = 2.8 mM^−1^ cm^−1^). The activity of APX is expressed as U mg^−1^ protein [59].

The nonspecific peroxidase (POD) activity was measured according to the method of Chance and Maehly [63]. The reaction mixture contained a 50 mM phosphate buffer (pH 7), 5 mM H_2_O_2_, 18 mM guaiacol, and a suitable enzyme extract aliquot. POD activity was estimated according to the increase in the absorbance of oxiguaiacol at 470 nm (ε = 26.6 mM^−1^ cm^−1^) and is expressed as U mg^−1^ protein [59].

### 3.7. Statistical Analysis

Data were analyzed using the STATISTICA software (version 10.0, StatSoft Inc., Tulsa, OK, USA, 2011). The results presented are the mean ± SD. Differences between the means were analyzed using an ANOVA followed by the post hoc Tukey’s test. A significant difference was considered at the level of *p* < 0.05. Prior to the statistical analysis, the normality of the data was evaluated using the Kolmogorov–Smirnov test implemented in STATISTICA. The *p*-values of the analyzed datasets were not significant, so the assumption of normality was accepted. The homogeneity of variance was assessed using the Levene’s test implemented in STATISTICA. The *p*-values of the analyzed datasets were higher than 0.05, indicating a homogeneity of variance. Data in the text, Tables and Figures were expressed as mean ± standard deviation (±SD), and error bars in the Figures indicate standard deviation. A principal component analysis (PCA) was carried out to identify the parameters that best describe the physiological performance of oxidative stress upon environmental conditions for the four tree species divided into defoliated (defoliation > 25%) and undefoliated trees using the STATISTICA software. Environmental data (i.e., ozone (O_3_), air temperature (T), air relative humidity (RH), solar radiation (Rad) and soil water content at different depths: SWC1 = 0–7 cm, SWC2 = 7–28 cm, SWC3 = 28–100 and SWC > 100 cm, and rain) were derived from ERA5 [64]. The variables were extracted from the gridded ERA5 dataset with a spatial resolution of about 33 km at the site location using bilinear interpolation.

## 4. Conclusions

This study focused on the oxidative stress indicators in four Mediterranean forest species. Considering the presence of ROS due to the stressful conditions present in the field, we found significant differences at the leaf level for all species between the defoliated and undefoliated trees for at least one of the examined parameters. After examining the biochemical indicators of oxidative stress, it is possible to determine whether a tree is in a state of stress before visual damage occurs. It was difficult to conclude which factors influenced the activation of the defense mechanisms of the examined species due to the small number of published studies carried out in the field and due to the many variables that can simultaneously affect a plant. Due to the small number of studies in the field and due to the many variables, that can simultaneously affect a plant, it is difficult to conclude which factors influenced the activation of the defense mechanisms of the examined species, and more studies like this are highly desirable. The long-term, field-based monitoring of environmental variables and forest health indicators at the same time could provide insight into the behavior of an individual species in response to external influences.

## Figures and Tables

**Figure 1 plants-11-03484-f001:**
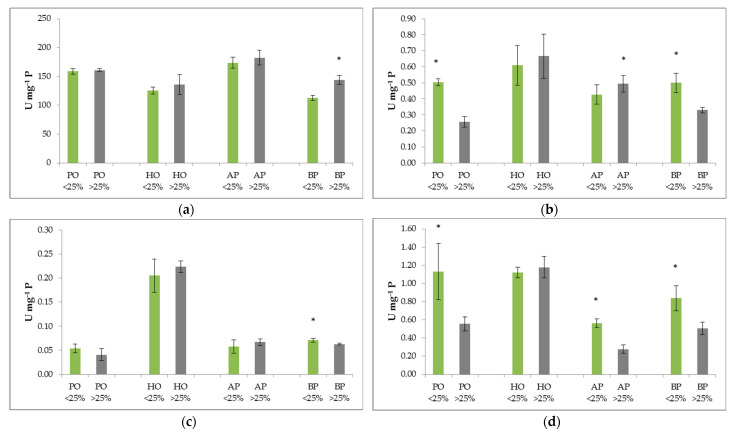
Activity of (**a**) superoxide dismutase (SOD), (**b**) catalase (CAT), (**c**) peroxidase (POD), and (**d**) ascorbate peroxidase (APX) in the leaves and needles of defoliated (defoliation of >25%, grey) and undefoliated (defoliation of ≤25%, green) pubescent oak (PO), holm oak (HO), Aleppo pine (AP), and black pine (BP) trees. Results are expressed as mean ± S. D. (*n* = 3). Values marked with a star (*) show a statistically significant difference between defoliated and undefoliated trees for a given species according to the Student’s *t*-test (* *p* < 0.05).

**Figure 2 plants-11-03484-f002:**
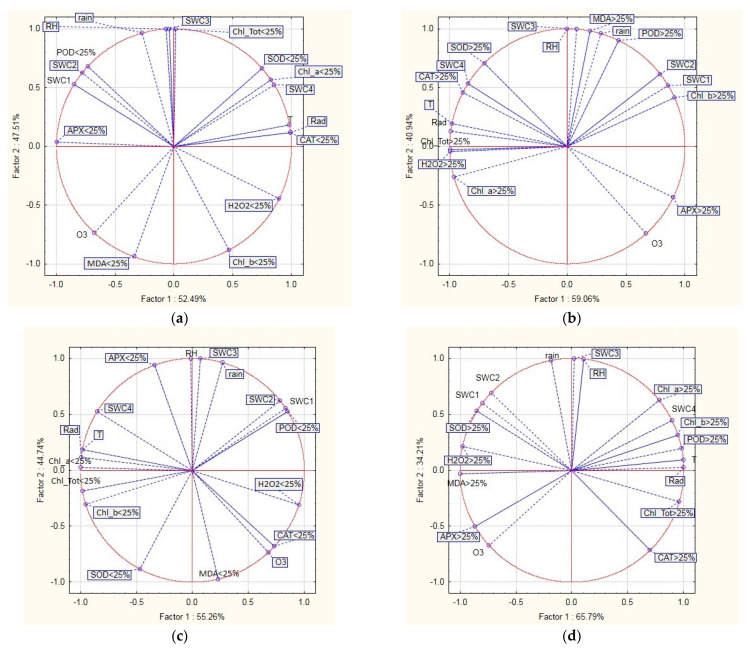
Principal component analysis (PCA) biplot of the physiological data and environmental variables for undefoliated (<25%) and defoliated (>25%) trees in four Mediterranean forest species: (**a**) pubescent oak (<25%); (**b**) pubescent oak (>25%); (**c**) holm oak (<25%); (**d**) holm oak (>25%); (**e**) Aleppo pine (<25%); (**f**) Aleppo pine (>25%); (**g**) black pine (<25%); (**h**) black pine (>25%). APX—ascorbate peroxidase; CAT—catalase; Chl_a—chlorophyll a; Chl_b—chlorophyll b; Chl_Tot—total chlorophyll; H_2_O_2_—hydrogen peroxide; MDA—malondialdehyde; O_3_—ground-level ozone; POD—peroxidase; Rad—solar radiation; rain—rain; RH—relative humidity; SOD—superoxide dismutase; SWC1—soil water content at 0–7 cm; SWC2—soil water content at 7–28 cm; SWC3—soil water content at 28–100 cm; SWC4—soil water content > 100 cm; T—temperature.

**Table 1 plants-11-03484-t001:** Concentrations of nutrients (N, P, K, Ca, and Mg) in the leaves and needles of pubescent oak, holm oak, Aleppo pine and black pine for defoliated and undefoliated trees. Results are expressed as mean values (*n* = 3). Values marked in bold show a statistically significant difference between defoliated and undefoliated trees for the same species according to the Student’s *t*-test (* *p* < 0.05). The colors indicate the concentrations of the elements: red—high; green—optimal; yellow—low [38].

Species	Category	N	P	K	Ca	Mg
		mg g^−1^	mg g^−1^	mg g^−1^	mg g^−1^	mg g^−1^
Pubescent oak	Undefoliated trees (≤25%)	16.95	1.22	6.62	10.72	1.23
	Defoliated trees (>25%)	16.95	**1.75 ***	**7.50 ***	**14.66 ***	**1.97 ***
Holm oak	Undefoliated trees (≤25%)	13.30	0.99	**6.26 ***	6.21	1.22
	Defoliated trees (>25%)	12.15	0.99	3.17	6.72	1.42
Aleppo pine	Undefoliated trees (≤25%)	11.14	1.34	3.99	**8.41 ***	1.65
	Defoliated trees (>25%)	10.88	1.33	3.96	4.03	1.70
Black pine	Undefoliated trees (≤25%)	10.11	1.20	4.98	**5.24 ***	1.30
	Defoliated trees (>25%)	9.69	**1.51 ***	**6.36 ***	2.52	1.22

**Table 2 plants-11-03484-t002:** Content of photosynthetic pigments in the leaves and needles (Chl-a, Chl-b, and Chl-tot), hydrogen peroxide (H_2_O_2_), and malondialdehyde (MDA) of pubescent oak, holm oak, Aleppo pine, and black pine for the defoliated and undefoliated trees. Results are expressed as mean ± SD (*n* = 3). Values marked in bold show a statistically significant difference between the defoliated and undefoliated trees of a plant species according to the Student’s *t*-test (* *p* < 0.05).

Species	Category	Chl-a	Chl-b	Chl-tot	H_2_O_2_	MDA
		µg g^−1^ FW	µg g^−1^ FW	µg g^−1^ FW	nmol g^−1^ FW	nmol g^−1^ FW
Pubescent oak	Undefoliated trees (≤25%)	**1503.93 ± 53.84 ***	366.39 ± 36.32	**1870.32 ± 90.16 ***	44.04 ± 12.88	**84.10 ± 7.33 ***
	Defoliated trees (>25%)	1302.68 ± 57.91	347.22 ± 35.22	1649.90 ± 93.13	39.4 ± 5.67	48.95 ± 4.13
Holm oak	Undefoliated trees (≤25%)	**1673.62 ± 69.47 ***	**466.84 ± 42.40 ***	**2140.46 ± 111.87 ***	0.03 ± 0.01	93.65 ± 7.57
	Defoliated trees (>25%)	1387.09 ± 55.64	357.96 ± 24.75	1745.05 ± 80.39	0.02 ± 0.00	**140.63 ± 6.16 ***
Aleppo pine	Undefoliated trees (≤25%)	406.74 ± 119.81	123.81 ± 25.77	530.55 ± 145.85	0.58 ± 0.09	16.49 ± 3.65
	Defoliated trees (>25%)	562.72 ± 104.72	**250.47 ± 34.61 ***	813.19 ± 139.33	0.61 ± 0.2	**31.60 ± 4.31 ***
Black pine	Undefoliated trees (≤25%)	666.30 ± 23.50	221.35 ± 13.81	887.65 ± 37.31	**0.89 ± 0.21 ***	27.76 ± 5.06
	Defoliated trees (>25%)	**795.59 ± 8.42 ***	241.20 ± 26.52	**1034.79 ± 34.94 ***	0.32 ± 0.06	24.57 ± 4.41

## Data Availability

Not applicable.

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
