# Peer review of "Are Foliar Nutrition Status and Indicators of Oxidative Stress Associated with Tree Defoliation of Four Mediterranean Forest Species?"

_plants, 2022, doi:10.3390/plants11243484_

Round 1

Reviewer 1 Report

The manuscript “Physiological Responses to Abiotic stress in four Mediterranean forest species” aim, in the words of the authors, “…to test whether foliar nutrition status and indicators of oxidative stress were associated with tree defoliation”. So, the title must be changed to focus the work presented, avoiding an expectation of a broader approach.

In my opinion, it is a manuscript needing an extensive editing of English language. Particularly in introduction, it is sometimes very hard to understand what the authors want to say. See, for example, the first phrases from introduction “Mediterranean region is center of biodiversity, but appears to be one of the most vulnerable regions and it is mainly caused by various human activities and climate change 34 [1]. The trend of increasing temperature, degradation of water resources and to the increased water demand and impacts of air pollution [2].”

Besides, introduction must be focused on the abiotic stress studied, foliar nutrition (even if the alteration in foliar nutrition is induced by others abiotic stresses), emphasizing the possible relation with defoliation via production of ROS. This must be also highlighted in discussion. For example, the authors began the discussion about chlorophyll content mentioning “Chlorophyll is one of stress indicators because its concentrations depend on various sources of stress, such as ozone, atmospheric and soil pollution [38,39] and stress intensity as well as duration [35,38].”, missing the well-known relation between chlorophyll content and foliar nutrition, mainly with the nitrogen.

Concerning results, the total chlorophyll content must be revised: it is impossible to obtain a lower value than the addition of chlorophyll a and b, as occurs sometimes in table 2(e.g., in Aleppo pine - Defoliated trees), and enzyme activities must be presented as U mg-1 protein or U mg-1 TSS (that is, total soluble protein).

Several other comments could be made after the suggested extensive revision of the English language, rewriting of introduction, carefully correction of the results, and a consequent discussion.

Author Response

Response to Reviewer 1 Comments

Comment 1: The manuscript “Physiological Responses to Abiotic stress in four Mediterranean forest species” aim, in the words of the authors, “…to test whether foliar nutrition status and indicators of oxidative stress were associated with tree defoliation”. So, the title must be changed to focus the work presented, avoiding an expectation of a broader approach.

Response: Thank you for the comments. The title has been changed to “Are foliar nutrition statues and indicators of oxidative stress associated with tree defoliation of four Mediterranean forest species? ” and now it is focused on the work presented. (title, p. 1, lines 2-3).

Comment 2: In my opinion, it is a manuscript needing an extensive editing of English language. Particularly in introduction, it is sometimes very hard to understand what the authors want to say. See, for example, the first phrases from introduction “Mediterranean region is center of biodiversity, but appears to be one of the most vulnerable regions and it is mainly caused by various human activities and climate change 34 [1]. The trend of increasing temperature, degradation of water resources and to the increased water demand and impacts of air pollution [2].”

Response: Thank you for the comment. Manuscript has been sent to Editing and Proofreading Services for an English language check.

Comment 3: Besides, introduction must be focused on the abiotic stress studied, foliar nutrition (even if the alteration in foliar nutrition is induced by others abiotic stresses), emphasizing the possible relation with defoliation via production of ROS. This must be also highlighted in discussion. For example, the authors began the discussion about chlorophyll content mentioning “Chlorophyll is one of stress indicators because its concentrations depend on various sources of stress, such as ozone, atmospheric and soil pollution [38,39] and stress intensity as well as duration [35,38].”, missing the well-known relation between chlorophyll content and foliar nutrition, mainly with the nitrogen.

Response: Thank you for the suggestion. The introduction has been rewrite as suggested. Also, results and discussion has been corrected according to your suggestions. (section 1, p. 2-3, lines 89-128 and section 2, p. 4-15, lines 177-567).

Comment 4: Concerning results, the total chlorophyll content must be revised: it is impossible to obtain a lower value than the addition of chlorophyll a and b, as occurs sometimes in table 2(e.g., in Aleppo pine - Defoliated trees), and enzyme activities must be presented as U mg-1 protein or U mg-1 TSS (that is, total soluble protein). Several other comments could be made after the suggested extensive revision of the English language, rewriting of introduction, carefully correction of the results, and a consequent discussion.

Response: Thank you for the comment. You are right about total chlorophyll content and it has been revised. Mistake in equation for Chl-tot calculation in excel sheet has been made. Values given in Table 2 have been corrected accordingly. (section 2, p. 7, lines 288-297). Also, units in the Figure 1 have been corrected (section 2, p. 10-11, lines 440-452).

Reviewer 2 Report

My comments are in the attached file.

Author Response

Response to Reviewer 2 Comments

Comment Admin 1:

1- English must revise carefully.

Response: Manuscript has been sent to Editing and Proofreading Services for an English language check.

2- Why the authors did not determine organic solutes and molecular parameters?

Response: This research was a part of the project that was dealing with this topic and due to financial restriction; we could not extend our research. We agree that it would be interesting to study suggested parameters.

3- Could the authors provide photos for the experiment?

Response: As mentioned in the manuscript (section: 3.1; line: 365), detailed description of plots can be found in:

  1. Jakovljević, T.; Lovreškov, L.; Jelić, G.; Anav, A.; Popa, I.; Fornasier, M.F.; Proietti, C.; Limić, I.; Butorac, L.; Vitale, M.; et al. Impact of Ground-Level Ozone on Mediterranean Forest Ecosystems Health. Sci. Total Environ. 2021, 783, 147063, doi:https://doi.org/10.1016/j.scitotenv.2021.147063.
  2. Jakovljević, T.; Marchetto, A.; Lovreškov, L.; Potočić, N.; Seletković, I.; Indir, K.; Jelić, G.; Butorac, L.; Zgrablić, Ž.; De Marco, A.; et al. Assessment of Atmospheric Deposition and Vitality Indicators in Mediterranean Forest Ecosystems. Sustainability 2019, 11, 6805, doi:10.3390/su11236805.

4- My other comments are in the text.

Response: Thank you for the comments. We corrected text as suggested

Reviewer 3 Report

Manuscript IDplants-2035167

Titled Physiological Responses to Abiotic stress in four Mediterranean forest species

Authors: Lovreškov et al.

The present manuscript by  Lovreškov et al describes the Physiological Responses of four Mediterranean forest species to the interactive effects of multiple abiotic stress. Although an array of analyses are presented, I do not believe that the degree of novelty justifies publication in Plants. In addition the  result and discussion secton is neraly discriptive and need intensif work to be rock solid.

In addition, the Results and discussion secton are very descriptif and need intensive work to be solid.

Also due to the analysis method used (ANOVA), it should be demonstrated that the plant samples belonging to the study groups meet the basic assumptions: normality, homogeneity of variance, and independence. There may be problems with the latter if the assessment of individual properties was carried out on samples from the same plants, i.e. for example, when determining the foliar nutritional status ​​and when determining the Activity of antioxidative enzymes, the same sample taken once from the groups (C, NPP, SPP) was used instead of a new, random sample being taken to examine each property. However, this requires many plants, which are not visible from the manuscript... But the other two assumptions must also be verified.

This is significant because the claims in the article are based on the results of this analysis, so this analysis must be rock solid.

Author Response

Response to Reviewer 3 Comments

Comment 1: The present manuscript by  Lovreškov et al describes the Physiological Responses of four Mediterranean forest species to the interactive effects of multiple abiotic stress. Although an array of analyses are presented, I do not believe that the degree of novelty justifies publication in Plants. In addition the  result and discussion secton is neraly discriptive and need intensif work to be rock solid.

In addition, the Results and discussion secton are very descriptif and need intensive work to be solid.

Response: Thank you for the comments. As concluded in our previously work (Jakovljević et al. 2021) the study was conducted on two Mediterranean species (Quercis pubescens and Pinus nigra) that were not been investigated yet. Therefore obtained results encourage us for further investigation presented in this work. Results and discussion have been rewritten as suggested (section 2, p. 4-15, lines 177-567).

Comment 2: Also due to the analysis method used (ANOVA), it should be demonstrated that the plant samples belonging to the study groups meet the basic assumptions: normality, homogeneity of variance, and independence. There may be problems with the latter if the assessment of individual properties was carried out on samples from the same plants, i.e. for example, when determining the foliar nutritional status and when determining the Activity of antioxidative enzymes, the same sample taken once from the groups (C, NPP, SPP) was used instead of a new, random sample being taken to examine each property. However, this requires many plants, which are not visible from the manuscript... But the other two assumptions must also be verified. This is significant because the claims in the article are based on the results of this analysis, so this analysis must be rock solid.

Response: Thank you for your comment. Explanation of ANOVA basic assumption evaluation was included into manuscript.

“Prior to statistical analysis normality of the data was evaluated using Kolmogorov–Smirnov test implemented in Statistica. p-values of the analyzed date sets were not significant, and therefore the assumption of normality can be accepted. Homogeneity of variance as assessed using Levene’s test implemented in Statistica. A p-value p-values of the analyzed date sets were higher that 0.05 indicating the homogeneity of the variances.“ (section 3.7, p. 17, line 669-674).

Trees were randomly selected and divided in two groups, as described in section 3.2. (Line: 581-592).

Experiment was designed and conducted according to UNECE ICP Forest manuals on sampling and analyses (References 18 and 56).

Round 2

Reviewer 1 Report

Only a last suggestion:

in line 29 "... whether a trees under stress even ..." should be " ... whether the trees are under stress even ...

Reviewer 2 Report

Accept in present form

Reviewer 3 Report

Dear Authors,

Thank you for your detailed answers. I accept your modifications.

With kind regards